# Mechatronic Design and Experimental Research of an Automated Photogrammetry-Based Human Body Scanner

**DOI:** 10.3390/s23135840

**Published:** 2023-06-23

**Authors:** Maciej Trojnacki, Przemysław Dąbek, Piotr Jaroszek

**Affiliations:** 1EDUROCO sp. z o.o., ul. Łąkowa 3/5, 90-562 Łódź, Poland; pjaroszek@eduroco.pl; 2ŁUKASIEWICZ Research Network—Industrial Research Institute for Automation and Measurements PIAP, Al. Jerozolimskie 202, 02-486 Warsaw, Poland

**Keywords:** 3D scanner, human body scanner, HUBO scanning system, photogrammetry, mechatronic design, laboratory studies, experimental research

## Abstract

The article concerns the mechatronic design and experimental investigations of the HUBO automated human body scanning system. Functional problems that should be solved by using the developed scanning system are defined. These include reducing the number of sensors used while eliminating the need to rotate a human and ensuring the automation of the scanning process. Research problems that should be the subject of experimental research are defined. The current state of the art in the field of systems and techniques for scanning the human figure is described. The functional and technical assumptions for the HUBO scanning system are formulated. The mechanical design of the scanner, the hardware and information system architectures as well as the user’s mobile application are presented. The method of operation of the scanning system and its innovative features are discussed. It is demonstrated that the developed solution of the scanning system allows the identified problems to be overcome. The methodology of the experimental research of the scanning system based on the photogrammetry technique is described. The results of laboratory studies with the use of dummies and experimental research with human participation are presented. The scope of the research carried out allows answers to the identified research problems related to the scanning of the human figure using the photogrammetry technique to be obtained. As part of laboratory tests using a measuring dummy, a mean error of 0.65 mm and standard deviation of the mean of 0.65 mm were obtained for the best scanner configuration. Research with human participation was carried out for the scanner version, in which the scanning time was 30 s, with the possibility of its reduction to 15 s. The results of studies using realistic dummies and with human participation were compared using the root mean square error parameter (RMSE) provided by the AliceVision framework, which was available for all analyzed objects. As a result, it was observed that these results are comparable, i.e., the RMSE parameter is equal to about 1 px.

## 1. Introduction

Full-body scanners are a special type of scanner, dedicated to the reconstruction of the human body as a whole, which enables, among other things, determination of the geometric parameters of the human figure based on the obtained 3D model. A list of such solutions can be found on the website [1], where out of the current 44 full-body scanner solutions less than half are currently for sale. In turn, the internet publication [2] made a subjective selection of the six best scanning systems currently available on the market, from the cheapest, at $10,000, to the most expensive, at $250,000.

It can be seen that, despite the market interest and solutions available, figure scanners are still not widely available in businesses where they could be widely used. First of all, virtual fitting rooms (VFRs) can be mentioned here, which allow the selection of clothes based on the actual dimensions of the human figure, determined from the 3D model. In turn, as indicated in the article [3], “despite the great potential of VFRs, their adoption is still in the preliminary stage”. Therefore, it can be said that the development of body scanning systems is still a current topic, and the possibility of implementing such solutions may be determined primarily by their price. Similar areas of application of body scanners may be tailoring or the modeling markets.

Another large industry in which body scanners can be implemented is medicine. In this application, 3D scanners can be used to scan selected body parts or the entire body. They can be used, for example, to support the therapy of posture defects as a measuring tool or in plastic surgery to visualize the current figure and present the possibilities of its correction. Body scanners can also be used to assess the effects of fitness training.

Body scanners can also be used in the entertainment industry, e.g., during events, and in the education industry, e.g., to popularize science and scanning techniques.

When analyzing the available solutions of automated scanning of the human figure, it can be noticed that there are three methods used to create a three-dimensional model of the object, according to which:the scanned person remains stationary and the device performs a series of measurements using a number of sensors permanently mounted in the space around the person,the scanned person is turned on a rotating platform and body measurements are taken from one side with the use of several sensors,the scanned person stands on a stationary platform, and the measurement is carried out using several sensors mounted on one or more arms rotating around the platform.

In the first method, i.e., the use of many sensors permanently mounted around the scanned person, a very short time of data collection can be obtained. However, this leads to a high price of the solution due to the cost of numerous sensors. This is a major disadvantage of this type of solution, because in order to perform an accurate reconstruction of the object, it may be necessary to use, for example, 200 cameras, in various poses in relation to the person.

As for the other two methods, which are related to movement of human or sensors, it can be seen from the review of existing solutions that systems with a rotating platform definitely dominate the market. However, the question arises as to which of these methods is more beneficial. To answer this question, two aspects can be taken into account—one related to usability and the other to technology.

From the point of view of the usability aspect, turning a person seems less advantageous, because a person involuntarily performs balancing body movements during rotation of the platform. This effect is the more noticeable the higher the velocity of rotation of the person. In addition, some people may feel discomfort during rotation, especially if the scan involves several turns.

Taking into account the technological aspect, the use of a rotating platform is simpler in terms of design. The platform can be a purely mechanical element, so there is no problem with powering the scanner. In the case of a rotating arm, on the other hand, power must be supplied to the sensors and other systems mounted on it. In the case of a rotating platform, however, a more powerful motor must be used, as it has to rotate the platform with a human.

Thus, it can be concluded that despite the less favorable solutions with a rotating platform, they are much more often used due to the simplicity of the construction. However, they require more motor power to rotate the human platform than to rotate the arm, so they may be less energy-efficient. 

The EDUROCO company took up the challenge of developing more functional and environmentally friendly scanners. The company is working on the development of solutions within two main product lines: portable scanners and scanbooths. In addition, the company is working on developing additional software modules for these scanning systems, dedicated to various applications, such as the clothing industry.

The solution to which this article relates allows the following main functional problems within the scanning system being developed to be overcome:limiting the number of sensors used for scanning, in relation to solutions in which they are mounted in fixed poses in relation to the scanned object, which should allow for a lower cost of the system production,scanning without rotating the person thanks to the scanner solution with a fixed platform and a mast rotating around it,ensuring the automation of the scanning process in relation to handheld scanners, which require the involvement of an additional person.

The scanning system is assumed to be a universal solution, allowing the use of various types of sensors and scanning techniques. This paper focuses on the version of the system that uses the photogrammetry technique, the goal of which is to obtain information about the physical environment from RGB images [4].

In terms of scientific contribution, this work is an attempt to answer, among others, the following research questions:How may the scanning accuracy be assessed?How to compare a 3D model with a higher-accuracy reference model?What is the reproducibility of the results in laboratory tests using a measuring dummy?What is the correlation between quality rates and the accuracy of body reconstruction at the stage of laboratory tests and in real conditions with the participation of humans?What are the differences and issues with scanning dummies and humans?What is the correlation between the results of quantitative and qualitative research?What are the results for different strategies in terms of camera positioning relative to the mast and the number of cameras used on the mast?What results are obtained depending on the height of the scanned person?What are the limitations of the photogrammetry in relation to scanning humans?

The above-mentioned research issues are general; therefore, research related to them can be carried out for various scanning systems.

Section 2 discusses the state of the art in the field of body scanning systems, taking into account various industries and technologies and indicates existing gaps. Section 3 shows the solution of the scanning system that solves the identified functional problems. In particular, it presents design assumptions, mechanical design of the scanner, mechatronic and software architecture, scanner operation and user application as well as innovative features of the system, which was finally achieved. Section 4 shows the results of laboratory research using dummies. Section 5 presents the results of experimental research in the field of human scanning. Section 4 and Section 5 focus on the research issues discussed above and constitute the main scientific contribution of the article. Section 6 summarizes the achievements in the field of the developed scanning system and the research results and indicates directions for further work.

## 2. State of the Art Review

The current state of the art in the field of human body scanning systems can be analyzed taking into account applications in various industries and the scanning techniques.

When it comes to virtual fitting rooms, in work [5], research was carried out with the use of four scanning systems based on various technologies in order to create virtual avatars for the selection of clothes. The analyzed solutions included: TG3D Scanatic 360 Body Scanner operating in infrared, Vitus Anthroscan by Avalution GmbH using laser scanning, as well as Ditus Smart also by Avalution GmbH and the Styku scanner by Styku, both based on the use of a Kinect sensor. Selected issues concerning the software enabling the selection of clothes were also discussed in that paper. 

As far as the medical industry is concerned, in [6], the use of a system based on photogrammetry for lower limb prostheses was considered. Article [7] discusses the possibility of design of patient-tailored assistive devices based on photogrammetric 3D body reconstruction. Publication [8] compares scanning systems based on photogrammetry and computed tomography in 3D medical measurements. Paper [9] discusses the possibility of using scanning systems to quantify aging-associated postural changes. Publication [10] uses a body scanner based on the RGB-D sensor and dedicated software tools to analyze spinal curves of patients based on 3D surface scans of their torsos. In turn, articles [11,12] discuss the possibility of creating anatomical models of the human body for medical education. 

In the field of fitness, the Naked scanning system [13], which is based on structured light technology, has been commercially available since 2017. From the same year, the Styku body scanner is also available, which is based on the use of the Microsoft Kinect V2 camera and photogrammetry technique, and enables the prediction of body fat [14]. In paper [15], that scanner was assessed in terms of what minimal change in the parameters of the human figure, including body fat, can be detected.

As for the state of the art in the field of body scanning systems, an up-to-date and comprehensive overview of the technology can be found in article [16], where these solutions were divided into three categories, including passive stereo, structured light and time-of-flight. As part of work [17], a comparison of photogrammetry and laser scanning techniques in application to medical education was made. It can be seen that each of the mentioned scanning techniques has its advantages and disadvantages, but what they have in common is the fact that only what the sensors see can be reproduced in the resulting 3D model.

Using photos from smartphone cameras and the photogrammetry technique, it is possible to obtain quite good-quality 3D models at a low cost, which results, for example, from the paper [18]. However, due to the complex reconstruction process, it is the most time-consuming of the mentioned techniques. Description of the state of knowledge in the field of photogrammetry can be found, among others, in [19,20]. Publication [21] compares affordable photogrammetry software for reconstructing a 3D model of a human foot, taking into account Agisoft Metashape, 3DF Zephyr and Regard 3D. In article [22], practical aspects of scanning the human figure with the use of a photogrammetry technique were discussed. Publication [23] presents the results of research on a system using 100 cameras and photogrammetry techniques for scanning the human figure. The paper [24] presents the results of research using multispectral full-body photogrammetry in the diagnosis of diseases.

It can be seen that the photogrammetry technique is used by as many as 16 solutions out of 44 listed on the website [1]. These include, among others, 3D Body Scanner by TWINSTER and 3D.me by Picanova. For some systems, information on the declared accuracy of reconstruction is provided. One of the highest accuracy, equal to 0.1 [mm], is provided by the myeggO Generator by tOOiin system [25].

Sensors using structured light allow for high-accuracy object reconstruction in a shorter time compared to the photogrammetry technique. Such systems are quite popular for human scanning, as the list of scanners at [1] contains information about 16 out of 44 solutions of this type. They include, e.g., Twinstant Mobile by Twindom and Proscanner by Fit3D. Unfortunately, information on the accuracy of the obtained model is available for only a few systems belonging to this group. However, from the comparative analysis presented in [26], no worse results can be expected to be obtained with this technique than from the photogrammetry technique. For example, the Twinstant Mobile by Twindom system, costing $26,995, allows a resolution of 0.7 [mm]. The results of research in the field of human scanning with the use of structured light sensors can be found, for example, in article [27]. In turn, ref. [28] presents a system using such sensors for measuring the bust.

Laser systems are relatively expensive, especially if the resulting model accuracy should be no worse than that obtained from a vision system. Therefore, the laser triangulation technique is used only by two scanners out of 44 solutions listed in [1]. They include Scanatic™ 360 Body Scanner by TG3D Studio and VITUS 3D body Scanner by Vitronic. Due to the fact that the point cloud is immediately available as a result of scanning, the data-processing time to obtain a 3D model can be much shorter compared to vision systems. 

Hybrid sensors which combine, for example, laser technology with vision technology, and which can also be used to scan the human figure, are also available. One example is the Intel^®^ RealSense™ LiDAR Camera L515. However, it can be said that the laser technique is dominant here, because as a result of scanning, a cloud of points from the LiDAR sensor is obtained, which is colored on the basis of information obtained from the RGB camera.

The possibility of fusing data from scanning systems using various techniques is discussed, e.g., in article [29], which shows the integration of laser scanning and photogrammetry techniques in the reconstruction of historic architectural objects. A similar approach was also used in publication [30] for statue scanning. Paper [31] presents the results of research in the field of reconstruction of small objects with the use of a scanning system constituting a combination of photogrammetry and photometric stereo. This allowed for a higher accuracy of the reconstruction compared to using a single solution. In turn, article [32] describes the use of a system integrating data from computed tomography and a system based on photogrammetry.

In [33], a comparison of the accuracy of body parameter measurement with the use of scanning systems and manual methods was made. It shows that, using the correct approach, one can achieve greater measurement accuracy using the analyzed automated systems. Article [34] is also devoted to a similar topic, in which, based on the analysis of available body scanning systems, it was found that they are a good technology for body assessments compared to manual measurement methods. In turn, in work [35], the results of body measurement using 3D body scanning mobile applications, 3D full-body scanning laser technology and the manual method were compared. As a result, it was found that even simple tools provide acceptable measurement accuracy and allow their use in certain industries.

There are also attempts to standardize the processing of 3D scanning results to ensure interoperability, i.e., to allow exchange of 3D human body anthropometric-related information between different systems. For this purpose, a working group was established and the results of the analyses in this area in 2019–2020 are presented in article [36].

Summing up the literature review, it can be seen that despite the large number of scanning systems for various industries, new solutions tailored to specific needs are still being developed. There are expensive systems on the market in the form of high-class scanbooths, in which there is usually a large number of permanently mounted cameras. There are also mid-class solutions that are transportable and have fewer sensors, typically mounted on the mast. In this case, scanners with a rotating platform on which a person is standing predominate. In turn, in the field of scanning techniques, solutions using structured light cameras or RGB cameras and the photogrammetry technique dominate.

As indicated in the introduction and developed in this literature review, mid-class solutions tend to suffer from human rotation, while high-quality solutions are very expensive due to the use of a large number of sensors. Therefore, it is advisable to develop compromise solutions that ensure the automation of the scanning process, a relatively short scanning time, typical for mid-class solutions, and the highest possible accuracy of object reconstruction.

## 3. HUBO Scanning System

### 3.1. Design Assumptions

The basic design assumption was to develop a portable and automated scanning system, characterized by a relatively high speed and accuracy of scanning, while maintaining a low production cost. It was assumed that the low cost of production will be ensured by using a small number of sensors in relation to solutions with permanently attached sensors around the scanned person. It was assumed that it would be implemented using a mast with sensors rotating around a stationary platform on which a person is standing. Additional features of the solution were, among others, the ability to configure the working space of the device to the assumed maximum dimensions of the scanned person, as well as the option of voice control of the scanning device and obtaining voice notifications about its work. The last feature was to facilitate self-scanning by the user and enable the use of the system by visually impaired persons.

The portability of the scanner was intended to enable its use in sectors where there is a need for frequent transport of the device, such as the entertainment or education sectors (e.g., educational workshops). This assumption also resulted in the requirements for quick folding and unfolding of the device for transport, ease of carrying, and modularity of the solution.

Scanning automation was to reduce the personnel required to operate the scanner and ensure the repeatability and accuracy of the obtained results, i.e., 3D models. Scanning with handheld scanners was the benchmark here.

The scanner was supposed to be a compromise between price and accuracy as well as scanning time. This means that it should be cheaper than expensive, fast and accurate systems, such as 3iosk by Mantis Vision [1], and at the same time allow faster and more accurate scanning than cheap manual scanners (there are available accurate high-class handheld scanners, such as 3D Einscan PRO HD, but their price is comparable to mid-class automated systems). 

### 3.2. Mechanical Design of the Scanner

The key functionality of the HUBO scanner is the automatic scanning of the human figure. For this purpose, the scanner has been designed so as to enable the collection of data on the scanned object, e.g., by taking a series of photos around it in the given camera poses. The design assumption was that it would be accomplished for a small number of cameras by using a fixed platform and a mast rotating around it. The mechanical design of the HUBO scanner operating under this assumption is illustrated in Figure 1.

In order to enable easy carrying and transportation of the device, it was divided into three modules: support module, platform module and mast module. These modules are connected with a minimum number of fasteners, also using wing screws to enable quick assembly and disassembly of the scanner.

The scanner construction uses trolleys, which can move in relation to the mast. Different types of sensors, such as RGB cameras or LiDARs, can be attached to the trolleys. The solution with trolleys allows the use of a small number of sensors compared to a solution in which the sensors would be permanently mounted on the mast. This also makes it possible to place the sensors in many positions in relation to the mast during scanning, which in combination with the rotational movement of the mast makes it possible to obtain many poses of the sensors in relation to the scanned object. Thanks to this solution, the scanner is also universal in terms of the number of camera poses, and at the stage of development of the scanner, it is easier to test its various configurations. In a situation where high accuracy is required, a large number of poses can be used, and when a short scanning time is required and accuracy is less important, a scan can be performed with a small number of poses.

### 3.3. Mechatronic Architecture of the Scanner

The scanner is a mechatronic device consisting of mechanical, electrical and electronic components, including programmable devices. For its correct operation, an information system is also necessary. The scanner mechatronic architecture is illustrated in Figure 2.

In order to supply individual subsystems, the scanner is equipped with a power subsystem, consisting of an AC adapter and DC converters of various voltages. 

The central part of the scanner is the control system, which includes computer resources responsible for data acquisition, coordinating the data collection process, and coordinating the operation of the entire scanner. The control system comprises a primary control subsystem, responsible for the operation of the entire scanner system, including the operation of sensors, and a secondary control subsystem, which implements controls commanded by the primary control subsystem. The secondary control subsystem is used to control the drive unit of the rotating arm, stepper motors to move the trolleys, servos to control the tilt of the sensors and LED lamps to set the correct light intensity.

The communication subsystem, in particular the wireless one, provides the control subsystem with access to the wide area network (WAN). The communication subsystem enables data exchange with the external parts of the information system, in particular the transfer of selected measurement data from sensors.

The sensor subsystem comprises sensors, which may be in particular cameras embedded in smartphones, depth cameras or LiDARs.

The arm-drive subsystem consists of a drive assembly (which includes a motor, a transmission and an encoder) and a magnetic field sensor, for resetting the angular position of the arm of the device. The arm-drive subsystem is responsible for the movement of the rotating arm of the scanner and is controlled by an arm-drive controller.

The trolleys’ motion subsystem serves to move the sensors along the arms of the mast. It consists of drive assemblies, which include a stepper motor and microswitches. 

The head-tilting subsystem is used to tilt the sensors by means of servo-drives, depending on their position along the length of the mast and the tilt angle of the mast itself. 

The lighting subsystem includes LED lamps for additional lighting of the scanned object.

### 3.4. Software Architecture of the HUBO Scanning System

The scanner information system includes control software implemented on the scanner and software external to the scanner itself, the architecture of which is shown in Figure 3.

The integral part of the external software architecture is the cloud information system, which provides, for example, the abilities to automatically obtain 3D models of scanned objects, to support multiple users with ensuring access security and to service a fleet of devices for scanning objects. This information system consists of the web application, computing application, database and data storage. These modules operate as part of the IT infrastructure that enables immediate adjustment of the supply of services to the demand of users. 

The web application is the link between all major components of the HUBO scanning system. The database stores the data necessary for the operation of web applications and computing servers, as well as data related to the scan results and target 3D models. The computing application is responsible for processing the data obtained during the scanning in order to obtain the final 3D models of the scanned objects. The data storage is responsible for long-term storage of binary files necessary for creating models and of selected multimedia files. 

The optional external services can include payment systems, social media systems, external 3D model provider systems, etc.

The architecture of the entire information system of the HUBO is distributed. The software also includes mechanisms that enable remote configuration and diagnostics of the scanner. In addition, processing in cloud computing to accelerate measurement data processing and to obtain a 3D model faster may be used as an option.

### 3.5. Scanner Operation and User’s Mobile Application

The operation of the scanner is based on the automatic collection of data on the scanned object in the given poses of the sensors in relation to it. For this purpose, various types of sensors can be used. In the case of the version of the solution used in this work, the scanner is equipped with smartphones that are used to take photos. These photos are then processed into a 3D model by the AliceVision framework [37], that is, using the photogrammetry technique, the mathematical foundations of which are described, among others, in in the article [4].

During scanning, the rotating arm turns alternately around the object. Before each turn of the arm, the trolleys are moved to the preset positions in relation to the mast and the sensors are tilted to preset pitch angles, which remain unchanged during the arm’s rotation. A single scan covers *n* positions of sensors around the object and *k* configurations in terms of position of trolleys and tilt of sensors. This gives a total of *n·k* data chunks from all sensors (photos in this case). If *m* trolleys with sensors are mounted on the scanner mast, then in order to obtain the dataset, including *k* configurations, it is necessary to perform *k*/*m* turns of the rotating arm. Thus, if *k* cameras were mounted on the mast, a single turn of the rotating arm would be enough to obtain an analogous dataset.

The paper analyzes a scanner solution with *m* = 2 trolleys, and the typical number of camera configurations when using the photogrammetry technique is *k* = 8, which requires four turns of the rotating arm to collect a dataset. By default, *n* = 32 photos are taken with each turn, resulting in a dataset of 256 photos.

The operation of the scanner is normally possible by means of a user’s mobile application (Figure 4). 

This application can optionally be voice-controlled and generate voice notifications about the scanning progress. After the scanning is complete and the dataset is converted into a 3D model, it is available in the application in the model’s section. 

### 3.6. Innovative Features of the HUBO Scanning System

The HUBO scanning system, in relation to the current state of technology, is distinguished by the following features present simultaneously in one solution:automation of the object-scanning process, enabling continuous operation of the scanner while reducing the necessary service by staff, or even eliminating it;greater scanning comfort for the user due to the fact that the platform does not rotate;enabling the configuration of the working space of the device for the assumed dimensions of the object;quick assembly and disassembly of the device for the purpose of its transport;a more energy-saving solution compared to a similar system with a rotating platform;lighting up the scanned object;providing access only to authorized users;ensuring the configuration of device parameters by the user in terms of scanning time and the resulting accuracy of the model;providing the ability of voice control of the scanner and receive voice notifications about its work;remote configuration and diagnostics of the scanner;processing in cloud computing to accelerate measurement data processing and obtain a 3D model faster.

## 4. Laboratory Research Using Dummies

### 4.1. Methodology of the Research

After the scanner demonstrator was designed, it was initially tested in laboratory conditions using dummies. At this stage, the following research problems were analyzed:A.selection of optimal geometrical parameters of the scanner;B.selection of the optimal distribution of cameras or other sensors and lamps;C.selection of optimal scanning parameters;D.selection of optimal settings for sensors and lamps;E.selection of optimal data-processing algorithms and their parameters.

The proposed research methodology, combined with the selection of scanner parameters, can also be successfully used in the process of designing other body scanners of a similar class.

The first two research problems, i.e., A and B, were the subject of an article [38] in which a method of selecting the geometrical parameters of the HUBO scanner was proposed. That method allowed a significant reduction in the number of selected parameters, because some of them are calculated on the basis of others. In addition, the derived dependencies made it possible to make the scanner parameters dependent on the height of the scanned person. Using that method, a comparative analysis of 3D models was made for various variants of the cameras’ distribution. The aim of the research at this stage was also to find the minimum number of camera poses to obtain a satisfactory 3D model, i.e., meeting certain quality criteria. Quality indicators were introduced to quantitatively assess the accuracy of the reconstruction and its effectiveness (the ratio of accuracy to processing time).

The research issues belonging to category C focused mainly on the selection of the type of movement of the rotating arm (continuous or intermittent) as well as the selection of optimal parameters of motion of this arm and the trolleys moving along the mast. Research in this area can be carried out similarly for other scanning systems that contain moving elements, e.g., a rotating platform.

Research in the field of object lighting, i.e., within the above-mentioned problems B and D, consisted in comparing the results for the case of using only external lighting, only lighting in the form of LED lamps mounted on the scanner and the use of both light sources. In addition, various settings of the LED lamps in terms of luminous intensity and color temperature were used in these studies. Research in the field of lighting is particularly important in the case of photogrammetry but should also be conducted for scanning systems using other sensors, such as depth cameras.

The research problems belonging to group E concerned data processing and included the pre-processing of photos before their further processing using photogrammetry algorithms, i.e., the use of various photogrammetry algorithms and their settings to obtain high-quality object reconstructions, as well as the processing of the obtained models to improve their quality (post-processing). If necessary, the post-processing of a 3D model can be done automatically using the filters usually implemented in the photogrammetry framework. For example, Alice Vision has smoothing and topology-preserving mesh regularization available directly in the model-generation pipeline. It is also possible to do a manual retouch of a 3D model using a general-purpose 3D editing software, such as open-source Blender.

A key element of the laboratory research was the qualitative and quantitative evaluation of the results. 

The qualitative assessment consisted of a visual evaluation of 3D models for various scanner configurations and their comparison with the real object and reference 3D model. The key criterion in this case was the completeness of the 3D model and the smoothness of its surface. Qualitative assessment was carried out at the initial stage of solution development, using a system demonstrator.

The quantitative assessment of the results consisted primarily in comparing the obtained 3D model with the reference 3D model. According to the assumptions of the project, the subject of scanning should be an object with a height of at least 150 cm and a width of at least 50 cm, for which a 3D model in the form of a point cloud is available. In addition, the requirement was that the reference 3D model should be developed for an object measured using a high-accuracy measuring system, i.e., not worse than 0.1 mm.

It was assumed that for the purpose of quantitative evaluation, the tested 3D model obtained from the HUBO scanner is transformed in order to compare in with a reference 3D model in such a way that the sum of squared errors in at least 100 characteristic points of these models is the smallest. This transformation should include translations and rotations as well as scale, if necessary. Then, one need to remove the so-called artifacts, i.e., points for which the absolute value of the error is greater than 5 times the standard deviation determined for all characteristic points. Finally, the accuracy of the model should be determined, in accordance with [39], as the A-type standard uncertainty calculated statistically from the set of observation values as the standard deviation of the mean, i.e., for all characteristic points, excluding the artifacts.

For this purpose, dedicated software has been developed that performs the described transformation and calculates statistical measures resulting from the comparison of 3D models for the analyzed measurement points (mean error and standard deviation of the mean). In the field of transformation of the tested 3D model to the reference 3D model, an effective method was implemented using the NVIDIA CUDA library.

### 4.2. Quantitative Research Using Reference 3D Model

In order to evaluate the scanning results, a measuring dummy was prepared, which was covered with a special pattern that can be described well with SIFT (scale-invariant feature transform) keypoints (see Figure 5a).

Then, the dummy prepared in this way was scanned using the Hexagon Stereo Scan (AICON) scanner measuring system (see Figure 5b), which has triangulation angle 27 deg, base length 470 mm, working distance 1000 mm, field of view 950 mm and a declared scanning accuracy of ±40 μm. In this way, a reference 3D model was obtained, which was then used to quantify the 3D models obtained for the tested HUBO scanning system, both using the demonstrator and the prototype. The obtained reference 3D model includes over two million points. 

During the laboratory research, the photogrammetry technique was used, thanks to which a cloud of points was obtained on the basis of a set of photos collected around the scanned object, and then a 3D model in the form of a triangle mesh. Figure 6 presents an example of the arrangement of cameras around the dummy during scanning and photos from selected camera poses. The determined camera distribution is the effect of the Structure-from-Motion (SfM) algorithm.

One of the issues that had to be solved as part of the quantitative research was how the measurement uncertainty for the obtained 3D models would be assessed in relation to the reference model.

Points in a dense cloud for the tested 3D model may have a distance greater than the measurement error. Thus, it would not be reasonable to determine the measurement uncertainty directly as the distance between the point $P_{i}$ of the reference 3D model and the nearest point $P_{j}$ of the tested 3D model.

Therefore, the measurement uncertainty for N≥100 measurement points was determined on the basis of the methodology described below.

Let us assume that:the measurement error is the distance of the point $P_{i}$ (for i=1, …, N) from the surface of the real object ξ;the measurement error for the reference 3D model at point $P_{i}$ is $dw_{i}$; the measurement errors for the tested 3D model obtained from the HUBO scanner at the nearby points $P_{j}$, $P_{k}$ and $P_{l}$ are $ds_{j}$, $ds_{k}$ and $ds_{l}$, respectively;in a typical case $ds_{j}\gg dw_{i}$, $ds_{k}\gg dw_{i}$ and $ds_{l}\gg dw_{i}$, where $dw_{i}\leq 0.04\;\mathrm{mm}\approx 0$.

In order to obtain a more adequate measure of accuracy, for each point $P_{i}$ from the reference 3D model, it is necessary to find the three points closest to it in the tested 3D model, which can be marked as $P_{j}$, $P_{k}$ and $P_{l}$ according to Figure 7. The points $P_{j}$, $P_{k}$ and $P_{l}$ form a triangle lying in the plane $\pi_{i}$.

Hence, the estimated measurement error of the HUBO scanner at the measurement point $P_{i}$, that is, $e_{i}$, was defined as the distance of point $P_{i}$ from the plane $\pi_{i}$. This is equivalent to finding a point $P_{m}$ on the surface $\pi_{i}$ and then the distance between the points $P_{i}$ and $P_{m}$.

In order to indicate that the three selected characteristic points are closest to the point $P_{i}$, for i=1, …, N, they will be further denoted as $P_{j}^{i}$, $P_{k}^{i}$ and $P_{l}^{i}$, whilst the point in the plane $\pi_{i}$ closest to the point $P_{i}$ will be denoted as $P_{m}^{i}$. 

The following formulas were used to determine the estimated measurement error $e_{i}$ for point $P_{i}$:

parameters of the general equation of the plane $\pi_{i}$, that is, $A_{i}x+B_{i}y+C_{i}z+D_{i}=0$, defined by the points $P_{j}^{i}$, $P_{k}^{i}$ and $P_{l}^{i}$:

(1)$${\left[A_{i},\;B_{i},C_{i}\right]}^{T}=\left(P_{k}^{i}-P_{j}^{i}\right)\times \left(P_{l}^{i}-P_{j}^{i}\right),$$(2)$${\left[A_{i},\;B_{i},C_{i}\right]}^{T}=a_{i}\times b_{i}={\left[a_{i2}b_{i3}-a_{i3}b_{i2},a_{i3}b_{i1}-a_{i1}b_{i3},a_{i1}b_{i2}-a_{i2}b_{i1}\right]}^{T},$$(3)$$D_{i}=-\left(A_{i}x_{j}^{\;}+B_{i}y_{j}^{\;}+C_{i}z_{j}^{\;}\right),$$
where $P_{j}^{i}$, $P_{k}^{i}$ and $P_{l}^{i}$ are position vectors of analyzed points,

estimated measurement error $e_{i}$ determined as a distance of point $P_{i}$ from the plane $\pi_{i}$, that is from point $P_{i}$ to point $P_{m}^{i}$:



(4)
$$e_{i}=\left|A_{i}x_{i}^{\;}+B_{i}y_{i}^{\;}+C_{i}z_{i}^{\;}+D_{i}\right|/\sqrt{{\left(A_{i}\right)}^{2}+{\left(B_{i}\right)}^{2}+{\left(C_{i}\right)}^{2}}=\left|P_{i}P_{m}^{i}\right|.$$



Finally, the measurement uncertainty of the HUBO scanner was calculated for all *N* measurement points as the standard deviation of the mean σ from the formulas [38]:(5)$$\sigma =\sqrt{\sum_{i=1}^{N}{\left(e_{i}-\mu \right)}^{2}/N},\;\;\;\;\mu =\sum_{i=1}^{N}e_{i}/N,$$
where μ is the mean calculated for all estimated measurement errors.

The proposed method of evaluating the accuracy of reconstruction can be used for any scanning system for which an accurate reference model is available.

Figure 8a shows a sample 3D model with texture obtained from the HUBO scanner, whilst Figure 8b shows the result of comparison of the reference 3D model with the 3D model from that scanner.

Quantitative research focusing on the selection of the geometrical parameters of the scanner and the arrangement of cameras around the scanned object (research problems from groups A and B) were the subject of publication [38]. They were related to the determination of the mean error μ for the measurement points in relation to the reference 3D model and the standard deviation of this mean σ. In addition, the study analyzed the value of the root mean square error (RMSE) parameter provided by the AliceVision framework for the Structure from Motion step. 

For the purposes of simulation studies for various variants of scanner settings, a theoretical parametric human figure in front and side views was developed separately for male and female. Such a parametric human figure makes it possible to visualize how the poses of the scanner cameras and the visibility of individual body parts would look for a selected human height and width. 

Figure 9a shows a theoretical human figure and the arrangement of cameras along the scanner mast (blue circles) for the best scanner configuration in terms of the accuracy of reconstruction among the analyzed variants, the direction of view of individual cameras (blue solid lines) and the range of their fields of view (red dashed lines).

The scanner is able to scan objects up to 2.2 m high, while the assumed minimum height is 1.0 m. The scanner ready for scanning has dimensions of about 1.74 m × 0.75 m × 2.45 m in standard configuration, and the scanning area is within a cylinder 2.45 m high and 1.75 m in diameter. 

It is true that the scanner can scan children shorter than 1 m; however, due to the problem with keeping a still pose by children, it was assumed that the minimum age of the scanned person is 5 years.

As a result of the tests for the configuration described above, the mean error for the measurement points was μ = 0.65 mm, the standard deviation σ = 0.65 mm, and the root mean square error parameter RMSE = 0.77 px. Moreover, the size of 1 px in the photo was estimated, which is equal to about 0.25 mm for the camera located at a distance of about 0.5 m from the scanned object. 

Figure 9b shows in the form of a boxplot the obtained RMSE quality rate values from seven trials for the configuration illustrated in Figure 9a. Based on these tests, the following values for this parameter were obtained: the lowest value 0.770 px, the highest value 0.816 px, the median 0.802 px, the arithmetic average 0.799 px and the standard deviation 0.015 px.

It was also noted that there is usually a correlation between the values of the RMSE, μ and σ parameters. In particular, for the best result regarding μ and σ, the smallest value of the RMSE parameter was also obtained.

The fact that there is a correlation between the values of the RMSE, μ and σ parameters, allows the use of the RMSE parameter during human scanning studies where, due to the lack of a reference 3D model, it is not possible to determine the accuracy of the reconstruction in the form of μ and σ measures. Moreover, the advantage of the RMSE parameter is that it is available already at the initial stage of processing, i.e., as a result of the Structure-from-Motion (SfM) algorithm.

The subject of laboratory research was also the analysis of the influence of the type of movement of the rotating arm and trolley on the obtained results (research problems belonging to category C). At the beginning, trials were carried out in the field of intermittent movement of the rotating arm. They consisted in stopping this arm each time after reaching a predetermined position in which pictures were taken by the cameras. This type of movement turned out to be impractical, as each stop caused the mast to swing. In this case, it was necessary to wait for the mast vibrations to cease before taking pictures, which made the scanning process take too long. Thus, the optimal solution was to take pictures during the rotation of the arm. Thanks to this, during the steady movement of this arm, there were no noticeable vibrations of the mast. However, there was a risk of blurring the photos due to movement of cameras. Therefore, the maximum angular velocity of the rotation arm at which this effect has not yet been observed was selected. On the other hand, in terms of the movement of the trolley, the optimal solution was one in which they are moved to new positions in relation to the mast between successive full rotations of the rotating arm. It was also assumed that, at the same time, the tilt of the cameras would also be changed using servos mounted on trolleys. Hence, during each full rotation of the pivot arm, the camera poses relative to the mast were constant.

As for the results of laboratory tests in the field of lighting (research problems from group D), it was noticed that a complete 3D model can be obtained without the use of LED lamps on the scanner, but their use increases the quality of the obtained 3D models. Therefore, usually the best results were achieved when intensive illumination of the scanned object was used, i.e., for the maximum power of the LED lamps on the scanner. In addition, the use of LED lamps allows for greater independence of the results from variable external lighting.

As previously mentioned, work on data processing (research problems from group E) included in fact pre-processing of photos and processing using photogrammetry technique. 

In terms of photogrammetry frameworks, the AliceVision framework was used. That framework allows good accuracy of the reconstruction to be obtained in a relatively short time. The use of various parameters of the AliceVision framework and their values was studied primarily to smooth the mesh surface of 3D models. As a result, a compromise setting was found, allowing on the one hand quite high fidelity of details to be obtained, and on the other hand to smooth the mesh surfaces of the 3D model. 

In the field of pre-processing, the impact of the effect of sharpening photos and taking photos with the use of the bokeh effect on the obtained 3D models was examined. However, the positive impact of these methods on the quality of the obtained 3D models was not noticed. For image processing using the photogrammetry technique, images in the original and reduced resolutions were taken into account. In the case of the AliceVision framework, the Downscale parameter of the DepthMap module was used for this purpose. It was noticed that in the case of using two-fold reduction of the photos in each of the two dimensions (i.e., from the resolution of 3016 px × 4032 px to 1508 px × 2016 px), the quality of the 3D models was similar to that of the original photos. However, this allowed for a significant reduction in processing time. 

However, in order to use 3D models in industries requiring a very accurate reconstruction of the object, additional post-processing may be necessary to improve these models.

### 4.3. Qualitative Research Using Realistic Dummies

An important element of laboratory research was also qualitative research using realistic dummies of various heights. For this purpose, a 1.86 m high male dummy (see Figure 10) and a 1.08 m high child dummy (see Figure 11) were used. 

During the research using these dummies, an analogous methodology was used in terms of scanner settings as in the case of the measuring dummy.

The preliminary experimental research revealed that the skin surfaces of the dummies are very smooth and have few features, which makes accurate reconstruction of these body parts difficult. On the other hand, human skin has many more visible features related to skin pores, wrinkles, fingerprint lines, hair, etc. Therefore, in order to reduce the differences in terms of visible features, additional features in the form of irregular marks of various sizes were applied to the surfaces of the hands and faces of the dummies, as shown in Figure 10b,c and Figure 11b,c. For the male dummy, tests were also carried out for the case of wearing woolen gloves on the hands. Figure 12 compares features for a human (male) hand, an original dummy, a dummy with additional features and a dummy in woolen gloves.

As part of the experimental studies for realistic dummies, the scanner setting strategies for which the best results were obtained during the research of the measuring dummy were first analyzed. 

This article uses the variant designation convention described in [38]. According to this convention, the four groups of research variants were distinguished, including:(a)the vertical distribution of camera viewpoints (on the z axis) which covered the variants: a1—even distribution of these points along the object height, a2—distribution of the viewpoints adjusted to the specificity of the figure;(b)the pitch angles of cameras, which were set so that for different variants there was a different number of viewpoints common for two cameras (one common viewpoint for variant b1 and three common viewpoints for variant b2); the distribution of viewpoints adjusted to the specificity of the figure was used for all variants in this group;(c)varied number of camera positions around the scanned object *n*, for the best variants of camera poses relative to the mast (variants from groups a and b),(d)varied number of camera positions along the mast *k*, where the starting points were the best strategies analyzed in the variants from groups a and b.

The best laboratory research results were obtained for variant b1, in which were used n=32 camera positions around the scanned object, k=8 camera positions relative to the mast and one viewpoint in the head area of the person, at which the two upper cameras were directed. Also promising was variant d2, in which a higher number of camera positions relative to the mast were used, i.e., k=10, the same number of camera positions around the scanned object, and in which there were three viewpoints in the upper part of the figure, where two cameras were directed at each of them.

Therefore, in the experimental studies presented in this paper, scanner settings similar to variants b1 and d2 were first used. Modifications have been made to these variants, including turning the smartphones upside down on the lower mast arm and optimizing the pitch angles of the individual cameras. Additionally, as part of this work, a new strategy was taken into account in terms of camera poses, i.e., eight camera positions along the mast were taken into account, but it was assumed that the mast will make two full rotations around the scanned person instead of one, and for each turn different pitch angles of cameras will be used. As a result, two times more camera poses were obtained in relation to the b1 variant, and this type of variant was called e1. This new variant can be used in the case of a scanner with cameras at fixed positions along the mast (i.e., in the version of the scanner without trolleys). This may allow obtaining a more accurate 3D model in relation to variant b1, but with twice the scanning time and significantly longer data-processing time.

The method developed in article [38] in the field of scanner parameter settings allows adjusting the settings to the height of the scanned object. Various improvements to this method have been made in this paper, including the assumption that the top cameras are not used when scanning people of shorter height, i.e., the determination of the 3D model is based on a smaller number of pictures. Due to the differences in the scanner settings for the male and child dummies, resulting from their different heights, individual test variants are marked with the suffix ‘m’ and ‘c’ for the male and child dummies, respectively.

Figure 13 illustrates the configuration of the scanner in terms of the distribution of its cameras for the male dummy. For all variants, *n* = 32 camera positions around the scanned object were assumed during one full rotation of the mast. For the b1m and e1m variants, *k* = 8 camera positions along the mast were introduced, while for the d2m variant, *k* = 10. For the e1m variant, *p* = 2 rotations of the mast were carried out, where different settings in terms of camera angles were used for each, while for the remaining variants, *p* = 1 rotation of the mast was performed. Therefore, the total number of camera poses *c* for individual variants was, respectively, b1m: 256, d2m: 320, e1m: 512. The settings in terms of the number of camera poses for the analyzed variants are presented in Table 1.

Figure 14 shows the obtained non-textured 3D models for these variants, together with the enlargement of the head and hands, i.e., critical parts of the body from the point of view of reconstruction of the human figure. Looking at the individual 3D models of the male dummy, at first glance there are no major differences between them. When analyzing the individual fragments of the models in more detail, it can be seen that the highest accuracy of reconstruction in the area of the dummy’s head was obtained for the b1m variant, and in the area of the hand for the e1m variant. After looking at the photos of the hands, it was noticed that the reason for the poorer quality of the reconstruction in some cases is the limited depth of field of the cameras or overexposure of the photos due to too intense lighting with LED lamps. This resulted in worse visibility of features on the surface of the hand. This indicates the need to differentiate lighting along the height of the figure, which can be taken into account in further research.

Figure 15 illustrates the configuration of the scanner in terms of the distribution of its cameras for the child dummy. In this case, analogous variants were used as in the case of scanning a male dummy. The settings regarding the number of camera poses for individual variants are summarized in Table 1.

In turn, Figure 16 shows the results of the reconstruction of the dummy’s figure for these variants together with the zoom of critical fragments. The obtained textured 3D models for individual variants, visible in the figure, indicate the good quality of the object reconstruction. By enlarging selected fragments of the 3D models, it can be seen that the best reconstruction quality was obtained for the e1c variant, which is especially visible on the surface of the cap.

Experimental research on realistic dummies focused on qualitative research. This was due to the fact that in this case it would be much more difficult to keep the object stationary due to the movement of limbs, clothes, hair, etc. Therefore, it was not reasonable to build a reference 3D model for each dummy and compare it with models from the scanner. In order to quantitatively compare the results, only the RMSE parameter was used, which does not require comparison with the reference model. Moreover, as mentioned earlier, this parameter is a measure of object reconstruction accuracy. As a result of scanning these dummies, the average values of this parameter were obtained for the tested dummies and the analyzed variants of the scanner’s settings, as shown in Table 1.

It can be seen that in this case there is no such good correlation between the value of the RMSE parameter and the accuracy of the reconstruction assessed as a result of qualitative tests, as was the case with the measuring dummy. This is probably due to the fact that the RMSE parameter has higher values as the number of cameras increases. Therefore, it is reliable to compare the results by this quality rate for the same number of camera poses used when scanning an object.

Laboratory investigations with the use of dummies allowed for the initial selection of the scanner configuration, including the parameters defining the arrangement of cameras around the scanned object, lighting settings and motion parameters of the rotating arm and trolleys. In addition, as part of these studies, optimal data-processing algorithms and their parameters were selected.

## 5. Experimental Studies Involving Humans

In experimental studies involving people, the best scanner settings selected during laboratory research with the use of dummies were verified in real conditions. Experimental studies were carried out for people of different height, taking into account children and adults, women and men.

In the field of clothes, attempts were made to use those for which the best results of reconstruction using the photogrammetry technique can be obtained. In the first place, clothes made of non-shiny materials, with a high-contrast mélange pattern, were chosen. In practice, such assumptions are met by, for example, clothing such as mélange tracksuit, jeans, mélange t-shirt.

During the experimental research with the participation of humans, a version of the scanner was used in which 10 smartphones were placed in fixed positions along the mast, with the possibility of adjusting their pitch angles to the height of the scanned person. This made it possible to perform the entire scan during one rotation of the mast and thus reduce the total scanning time. In this case, the scanner settings were used in accordance with the previously analyzed variant d2. 

The total time required to achieve a final 3D model can be divided into scanning time and data-processing time. The scanning time is related to the collection of the dataset, which is the focus of this article. The scanning time depends on the version of the scanner (with fixed cameras in relation to the mast or in the version with two cameras placed on mobile trolleys) and lasts from 15 s to 180 s. As part of the experimental study involving humans, a version of the scanner with fixed cameras was used. A conservative scanning time of approx. 30 s was adopted in order to prevent possible blurring of some images, i.e., to guarantee the best possible accuracy of reconstruction. The processing time depends on the computer hardware and the number of photos, which ranged from 192 to 512 depending on the scanner configuration and object height. The processing time ranged from several minutes if cloud computing services were used for processing, to several hours in the case of a standard PC.

Figure 17 shows the scanning process of a 1.82 m tall man (co-author of the paper) and obtained results, i.e.: camera poses determined by the SfM algorithm, solid model and final textured solid model, as well as enlargement of the model’s head.

In terms of camera distribution, the quite large distance between the location of adjacent cameras on the lower and upper arm of the mast may be noticed. This is due to the fact that the smartphones on the lower arm are mounted upside down to increase the lower visibility range, while the upper ones are mounted normally to achieve the maximum upper range. Individual cameras are directed at precisely defined areas of the body, similarly to the d2m variant for a male dummy (see Figure 13). In particular, they focus on the head and hands, i.e., those parts of the body that are the most difficult to reconstruct. Analyzing the reconstruction results, i.e., both the non-textured and the textured solid model, one can notice a very good quality of the model, which is particularly visible on the example of the head.

Figure 18 presents the results of analogous studies for a 1.74 m tall woman. Analyzing the obtained results, one can notice a generally very good quality of the reconstruction. Only in the case of the face was the surface smoothness of the 3D model not satisfactory, which was probably the result of a small number of features on the skin surface, which made the reconstruction process more difficult. Higher quality in this respect can be achieved by changing the parameters of the photogrammetric pipeline or by using additional filters at the post-processing stage.

Figure 19 illustrates the scanning process and the obtained results for a 1.4 m tall boy. In this case, due to the lower height of the person, photos from the eight lower cameras were used for the reconstruction, i.e., the two smartphones on top were omitted. The obtained results indicate a similar accuracy of the reconstruction as before. In this case as well, the most problematic reconstruction was that of the face.

As in the case of experimental research using realistic dummies, due to the impossibility of guaranteeing the stability of the object, studies involving human subjects focused on qualitative research. As part of the experimental studies, as a quantitative measure only the values of the RMSE parameter were determined. In the presented cases, the RMSE parameter amounted to 1.032 px for the man, 0.936 px for the woman and 1.119 px for the boy. The values of the RMSE parameter are therefore approx. 1 px and are close to the values obtained during experimental investigations using realistic dummies (see Table 1). They are also slightly higher (worse) than the values obtained for the measuring dummy for the best configuration (see Figure 9 and [38]). 

As a result of experimental studies involving humans of different heights, slight differences in the obtained reconstruction accuracy were observed, which are within the error limits of the scanning system. This was achieved by parameterizing the settings of cameras in relation to the height of the scanned person. In particular, the individual cameras adapt their pitch angles to the height of the person. The factor affecting the deterioration of accuracy may be a small number of features visible on the surface of the object or the presence of a large number of similar features next to each other and the occurrence of reflective surfaces. These are the limitations of the used photogrammetry technique.

The correlation between the value of the RMSE parameter for a given scanner configuration and the accuracy of the reconstruction was noticed. Therefore, on the basis of obtained research results involving humans, it can be expected that the accuracy of the reconstruction of scanned persons is close to the quality obtained for realistic dummies. It can therefore be concluded that as a result of experimental research using the HUBO scanning system and the photogrammetry technique, satisfactory results were achieved in terms of reconstruction accuracy, which allow the solution to be used in various industries. The obtained results in relation to experimental studies using dummies are also particularly good due to the fact that the subjects of scanning were people who during scanning may perform unintentional, involuntary body movements, decreasing the accuracy of the reconstruction of the figure. 

## 6. Conclusions and Directions of Further Works

Based on the design process and experimental investigations, the following conclusions can be pointed out.

The developed and verified solution allowed the identified problems occurring in other solutions to be overcome. In particular, it enables the reduction of the number of sensors necessary for scanning in relation to systems with permanently installed sensors, which results in a much lower solution cost. The disadvantage of many solutions of a similar class related to rotating a person was also eliminated. In addition, it allows one to automate the scanning process, eliminating the need to engage an additional person to operate it.The developed scanner solution meets all design assumptions, including the maximum height of the scanned person, scanning time, and the accuracy of reconstruction. The user’s mobile application allows the user to perform self-scanning. The configuration of the scanner can adjust to the height of the scanned person.The selection of appropriate scanning and data-processing parameters is a multi-criteria optimization problem. In this work, the focus was on the issue of selecting the optimal parameters of the scanning system as part of research using realistic dummies and with the participation of humans, taking into account their different heights. The proposed methodology for selecting the distribution and parameters of sensors can also be used for other scanning systems of a similar class.The article presents a method of evaluating the accuracy of reconstruction using a high-accuracy reference 3D model and based on selected measurement points. Since the accuracy assessment is independent of the scanner design and scanning technique, this approach can be successfully applied to any scanning system.Within the laboratory research using dummies and the experimental studies involving humans, a number of research problems that are within the scope of this work were also analyzed. As a result of quantitative and qualitative research, answers to key research questions were obtained and limitations of the photogrammetry technique in the application to scanning people were identified. The analyzed research issues are so general that research related to them can be conducted for various scanning systems. Moreover, most of them can be performed for the photogrammetry and other scanning techniques as well.The RMSE parameter was used to quantitatively assess the accuracy of the reconstruction in experimental studies, which can also be used in investigations of other scanners based on the photogrammetry technique.As a result of the research, it was noticed, among other things, that the photogrammetry technique does not cope well with objects that have few features, where many similar features are next to each other, or where there are shiny surfaces. Moreover, there are inaccuracies in the reconstruction in the case of blurred or overexposed photographs, which make it difficult to correctly find the features of the object.However, if reconstruction errors are encountered, one can try to combine the results for different processing parameters, also applying this treatment in an automatic way.As a result of experimental research, satisfactory results of reconstruction of human figure were obtained, in the cases of both scanning dummies and real people.Scanning people is more problematic than scanning dummies due to the occurrence of involuntary body movements that might decrease the accuracy of the reconstruction.Considering the achieved functionalities, the HUBO scanning system can already be used in selected industries, e.g., modeling.

Further research work may focus on the issues listed below.

The use of other types of sensors, such as depth cameras and related data-processing methods, which will enable processing the point cloud right away.The development of data-processing technique alternatives to photogrammetry, e.g., NeRF, which will enhance the accuracy of reconstruction and overcome the limitations of the photogrammetry technique related to object features, including shiny surfaces.The application of artificial intelligence methods to improve the accuracy of reconstruction, including the completion of missing fragments and smoothing of surfaces.Development of methods for determining anthropometric parameters of the human figure for use in various industries, e.g., clothing, fitness and medical.

## Figures and Tables

**Figure 1 sensors-23-05840-f001:**
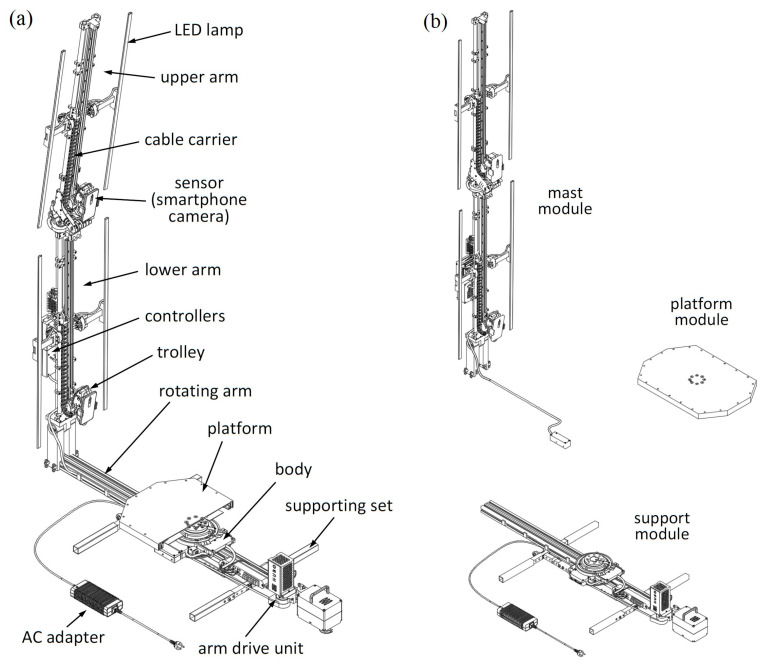
Illustration of the mechanical design of the HUBO scanner (**a**) and the division of the scanner into modules (**b**).

**Figure 2 sensors-23-05840-f002:**
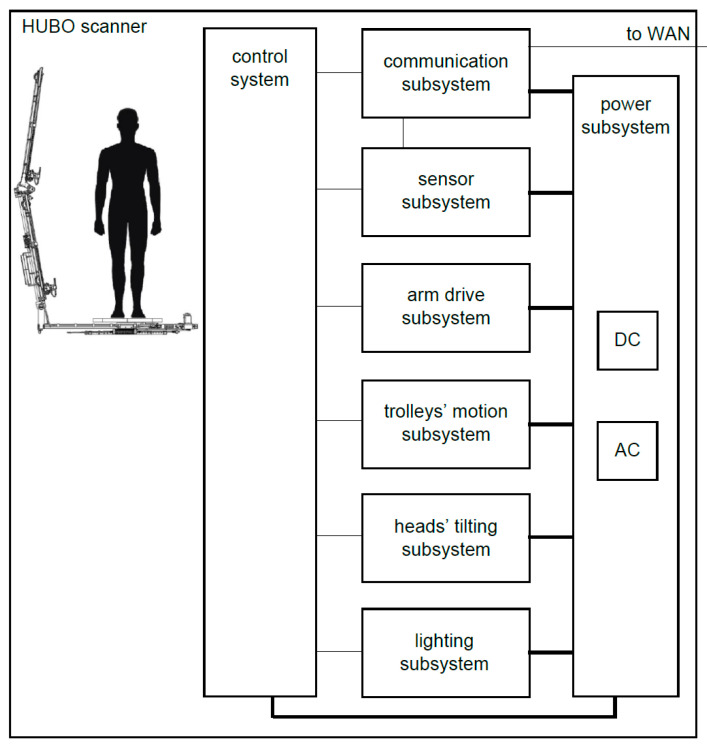
Mechatronic architecture of the scanner.

**Figure 3 sensors-23-05840-f003:**
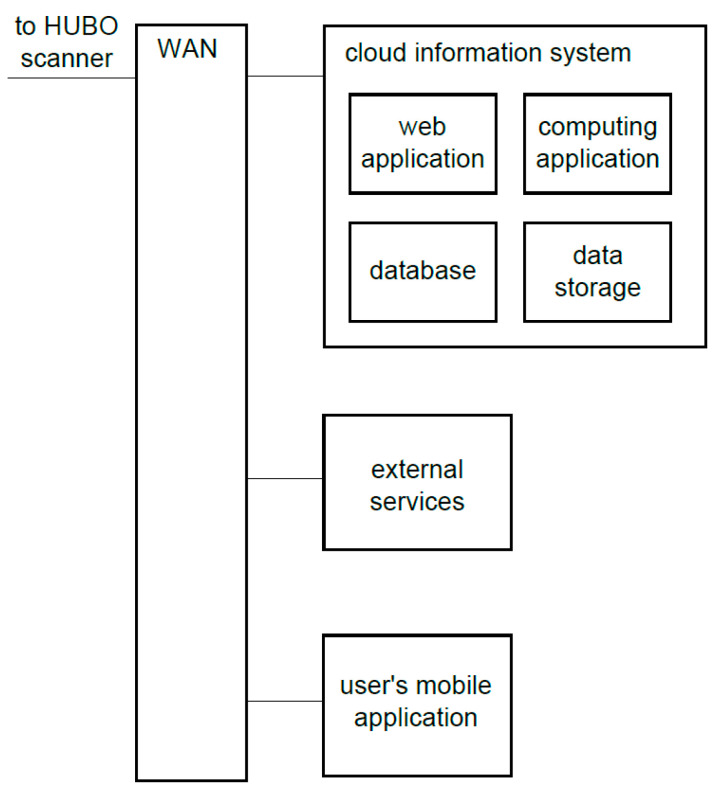
Software architecture of the HUBO system external to the scanner device.

**Figure 4 sensors-23-05840-f004:**
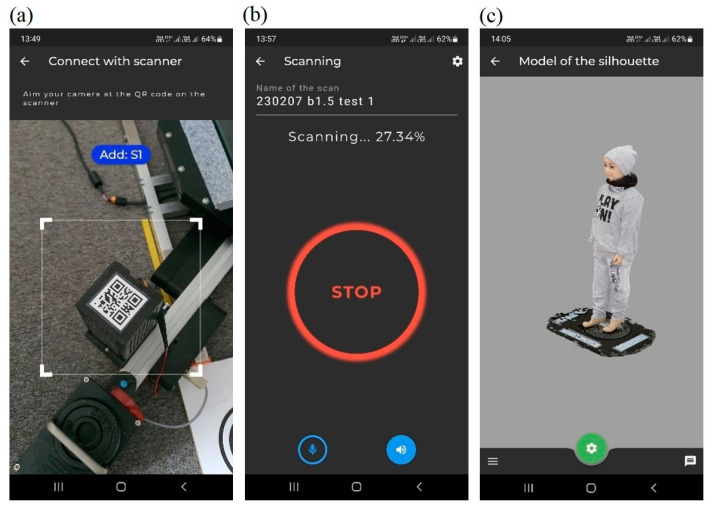
Mobile application of the scanner: scanner identification using a QR code and connection to it (**a**), scanning in progress (**b**), 3D model viewing (**c**).

**Figure 5 sensors-23-05840-f005:**
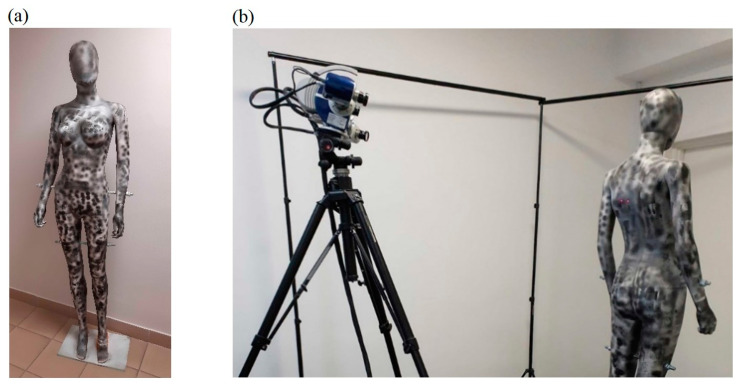
Specially prepared female dummy used during laboratory research of the HUBO scanning system (**a**) and scanning the measuring dummy to obtain a reference 3D model (**b**).

**Figure 6 sensors-23-05840-f006:**
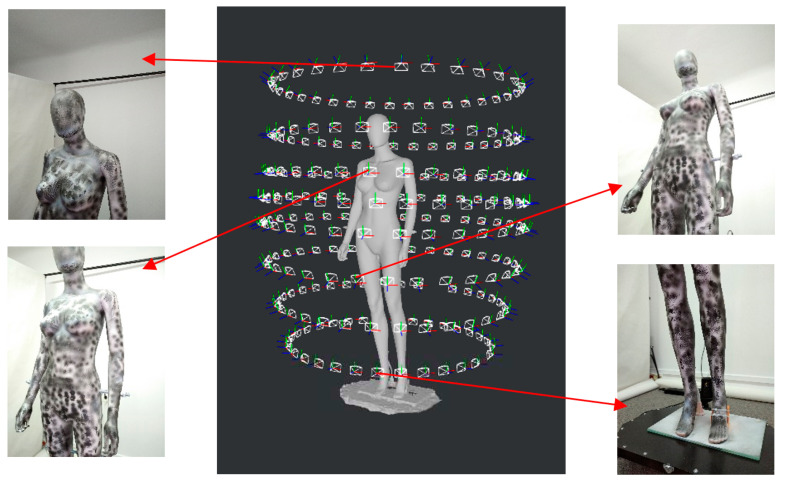
An example of the arrangement of cameras around the dummy during scanning and photos from selected camera poses.

**Figure 7 sensors-23-05840-f007:**
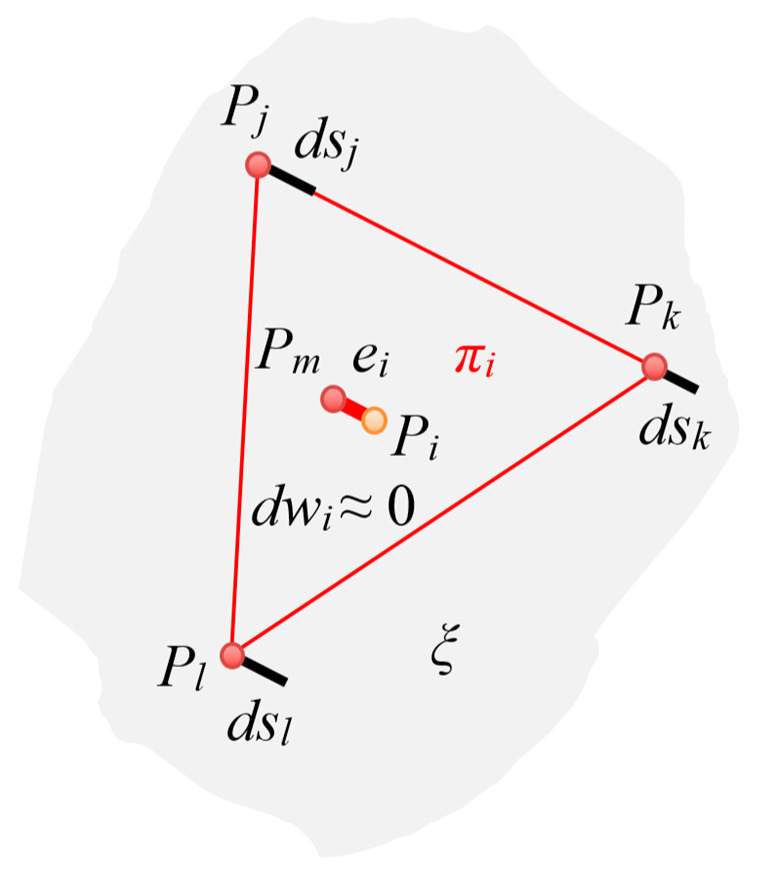
Illustration of the distribution of points for the reference and tested 3D model.

**Figure 8 sensors-23-05840-f008:**
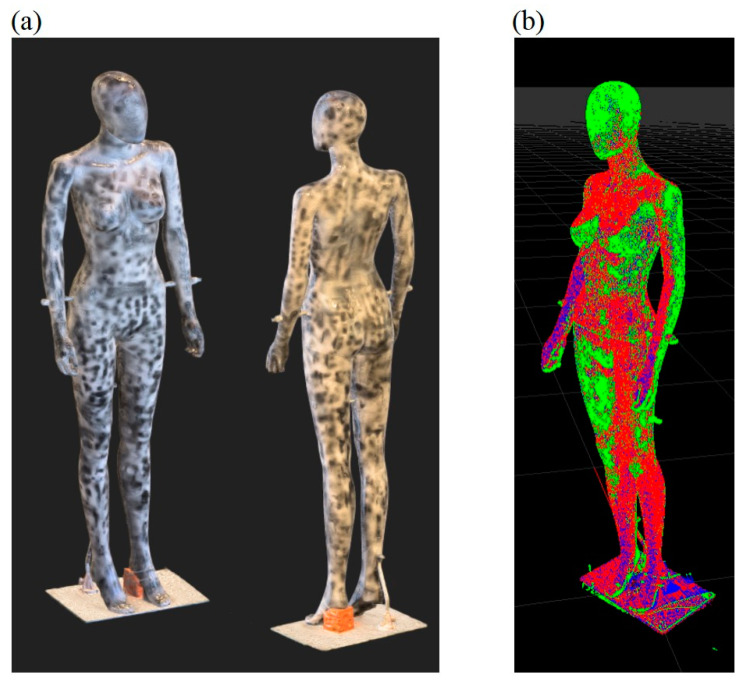
The results of the scanner accuracy research: sample 3D model obtained from the HUBO scanner (**a**), comparison of the reference 3D model (red) with the 3D model from the scanner (green) (**b**).

**Figure 9 sensors-23-05840-f009:**
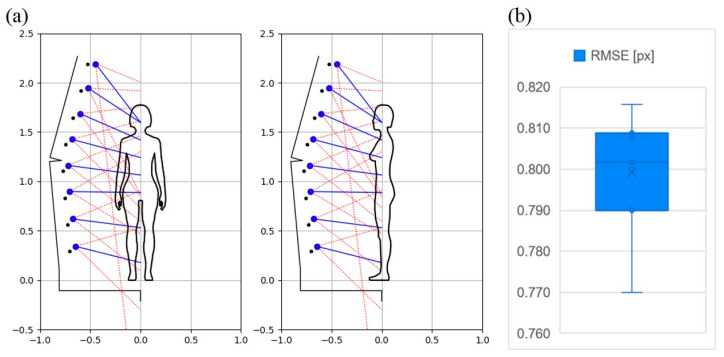
Illustration of the arrangement of cameras along the scanner mast for the best-found configuration of the scanner due to the accuracy of the reconstruction (**a**), the results of quantitative research in terms of the RMSE quality rate for that configuration (**b**).

**Figure 10 sensors-23-05840-f010:**
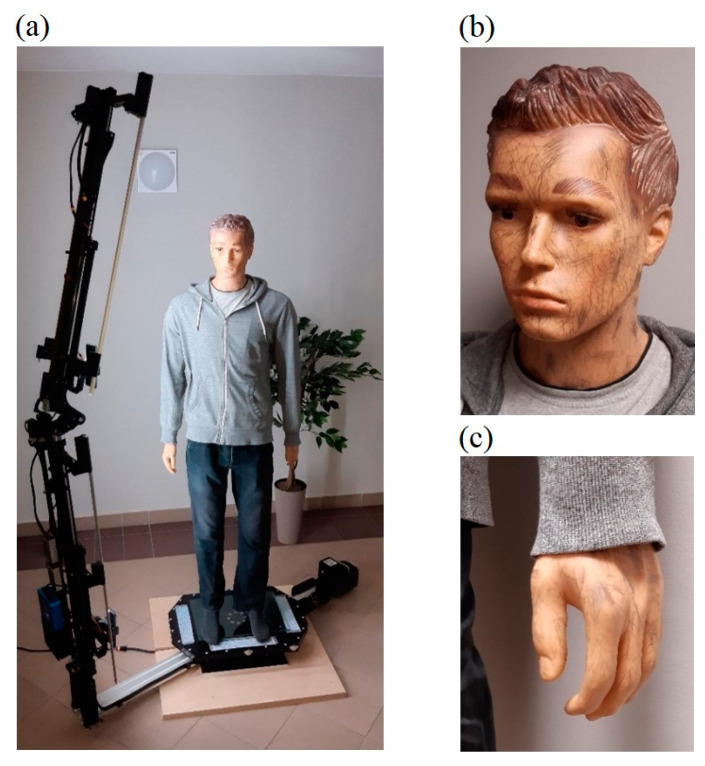
Male dummy with a height of 1.86 m (**a**), added features to the skin of the face and hands (**b**,**c**).

**Figure 11 sensors-23-05840-f011:**
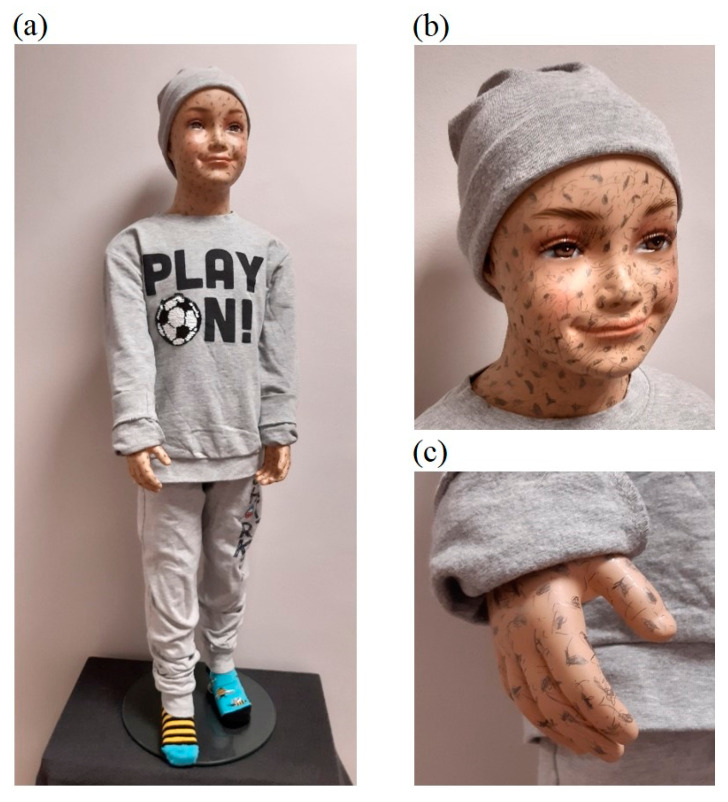
Child dummy with a height of 1.08 m (**a**), added features to the skin of the face and hands (**b**,**c**).

**Figure 12 sensors-23-05840-f012:**
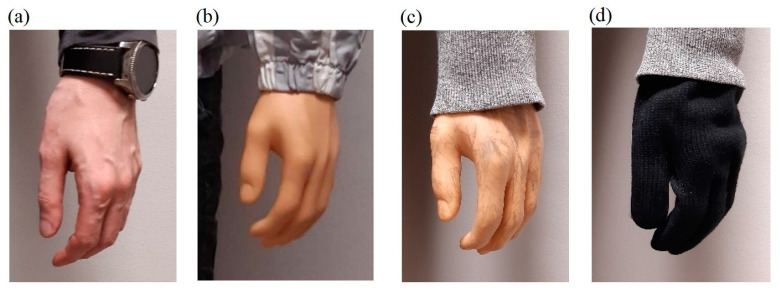
Comparison of hand features: human (**a**), dummy (**b**), dummy with added features (**c**) and dummy in woolen gloves (**d**).

**Figure 13 sensors-23-05840-f013:**
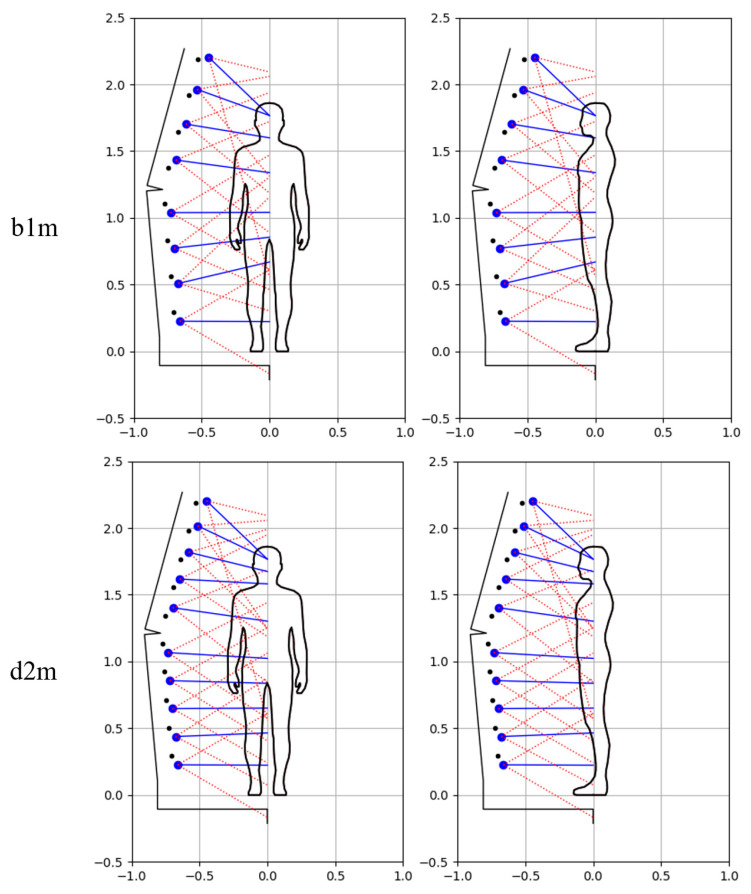
Illustration of the arrangement of cameras along the scanner mast for male dummy and analyzed variants.

**Figure 14 sensors-23-05840-f014:**
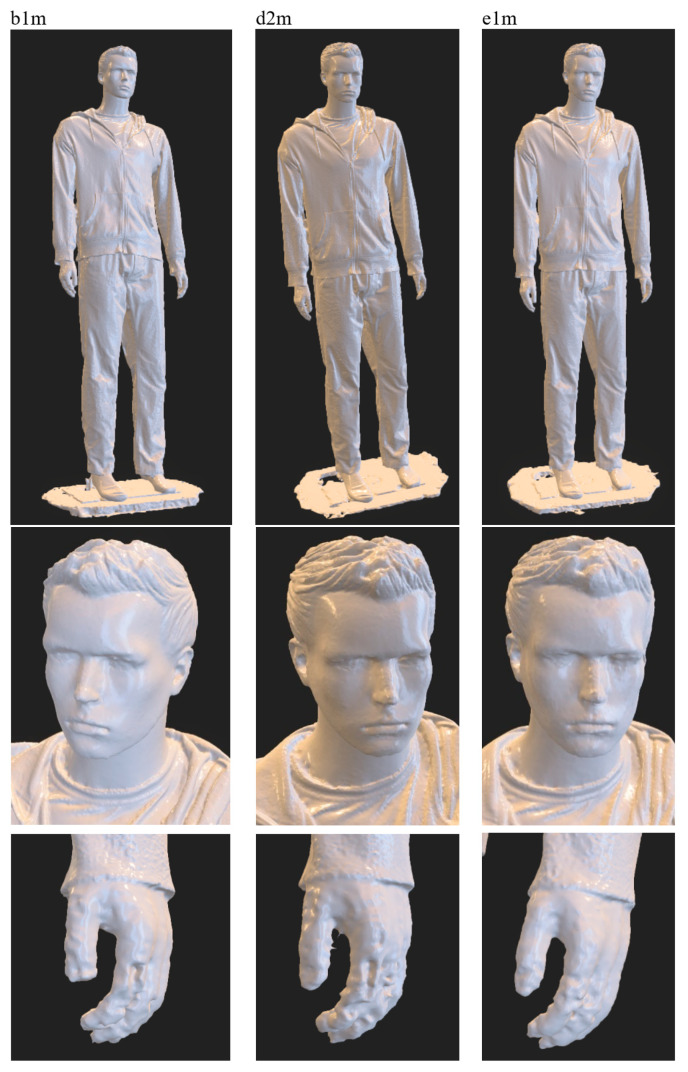
Exemplary 3D model of a male dummy obtained within laboratory research.

**Figure 15 sensors-23-05840-f015:**
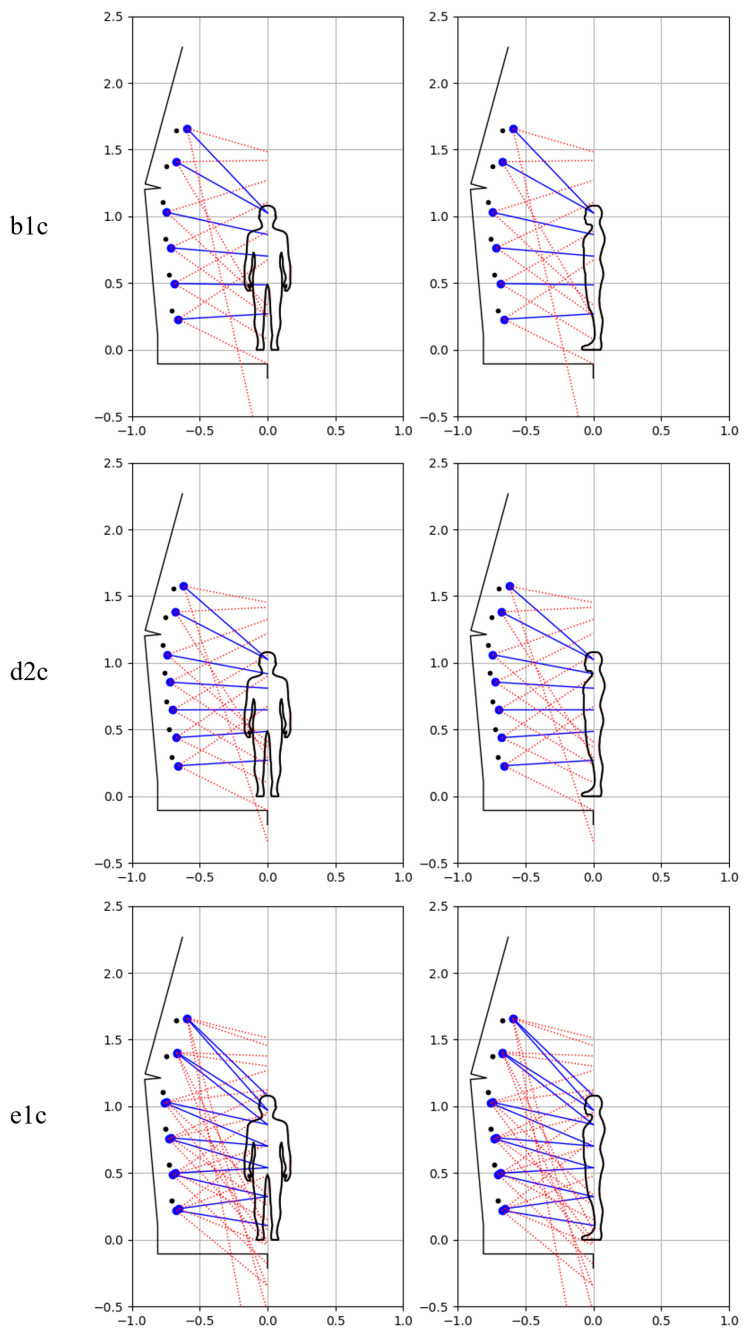
Illustration of the arrangement of cameras along the scanner mast for child dummy and analyzed variants.

**Figure 16 sensors-23-05840-f016:**
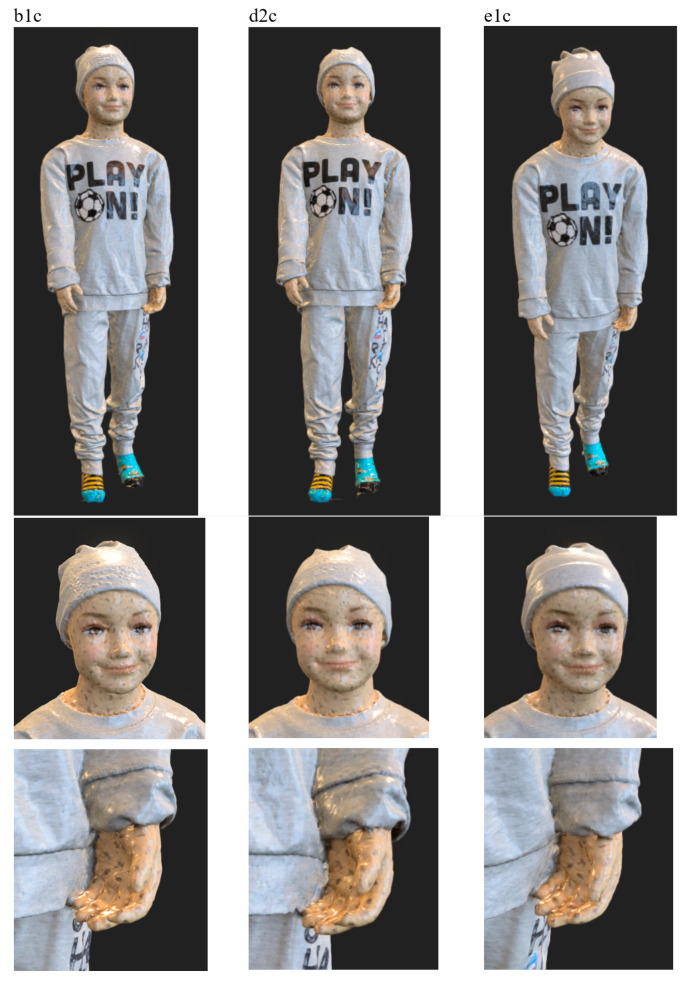
Exemplary 3D model of a child dummy obtained within laboratory research.

**Figure 17 sensors-23-05840-f017:**
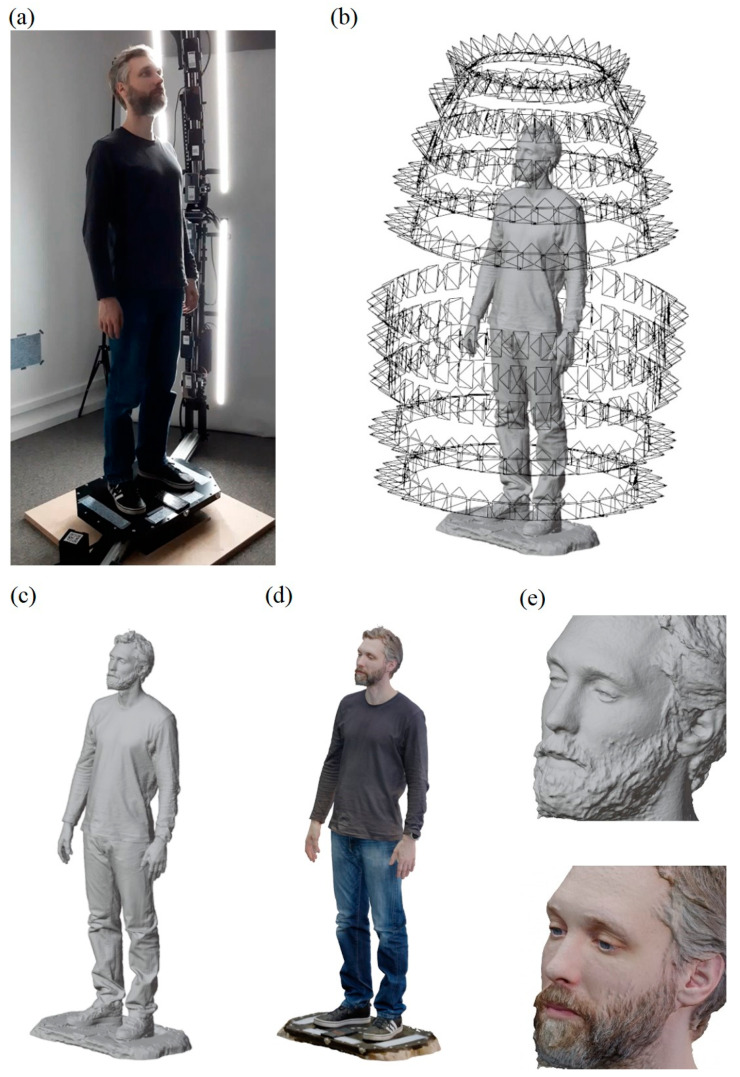
Scanning of a 1.82 m tall man using the HUBO scanner (**a**), determined camera poses (**b**), non-textured solid model (**c**), textured solid model (**d**), enlargement of the model’s head (**e**).

**Figure 18 sensors-23-05840-f018:**
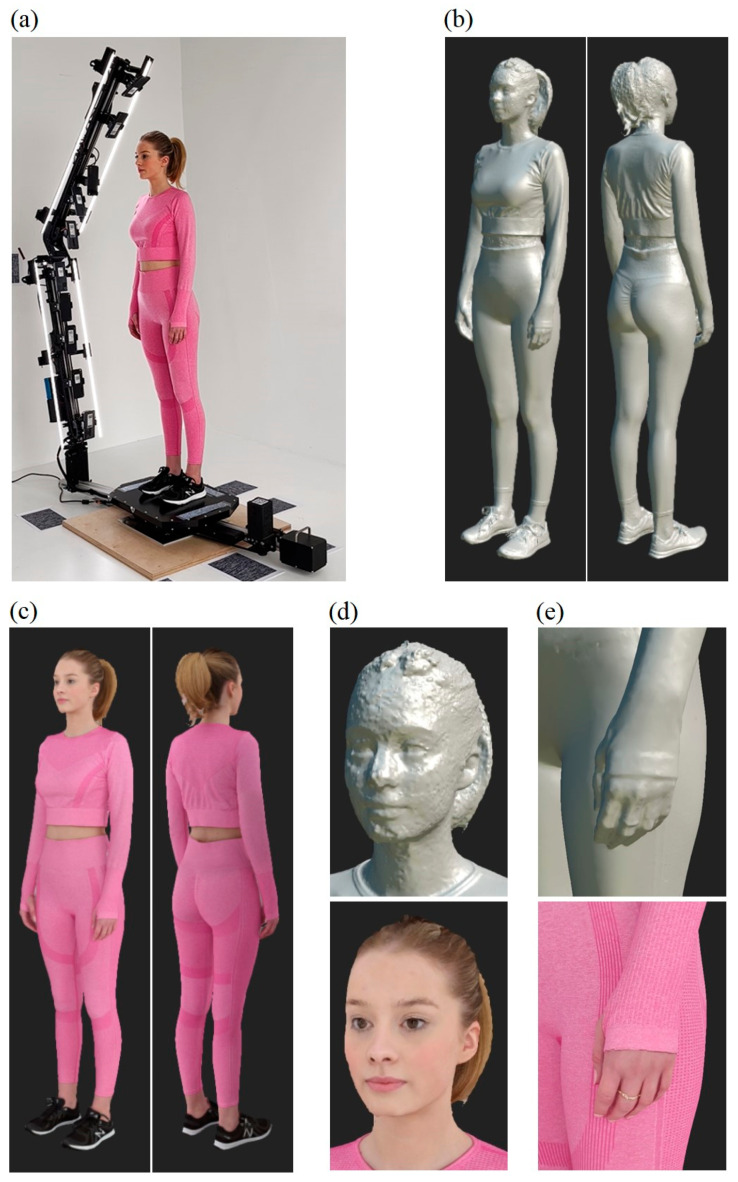
Scanning of a 1.74 m tall woman using the HUBO scanner (**a**), non-textured solid model (**b**), textured solid model (**c**), enlargement of the model’s head and hand (**d**,**e**).

**Figure 19 sensors-23-05840-f019:**
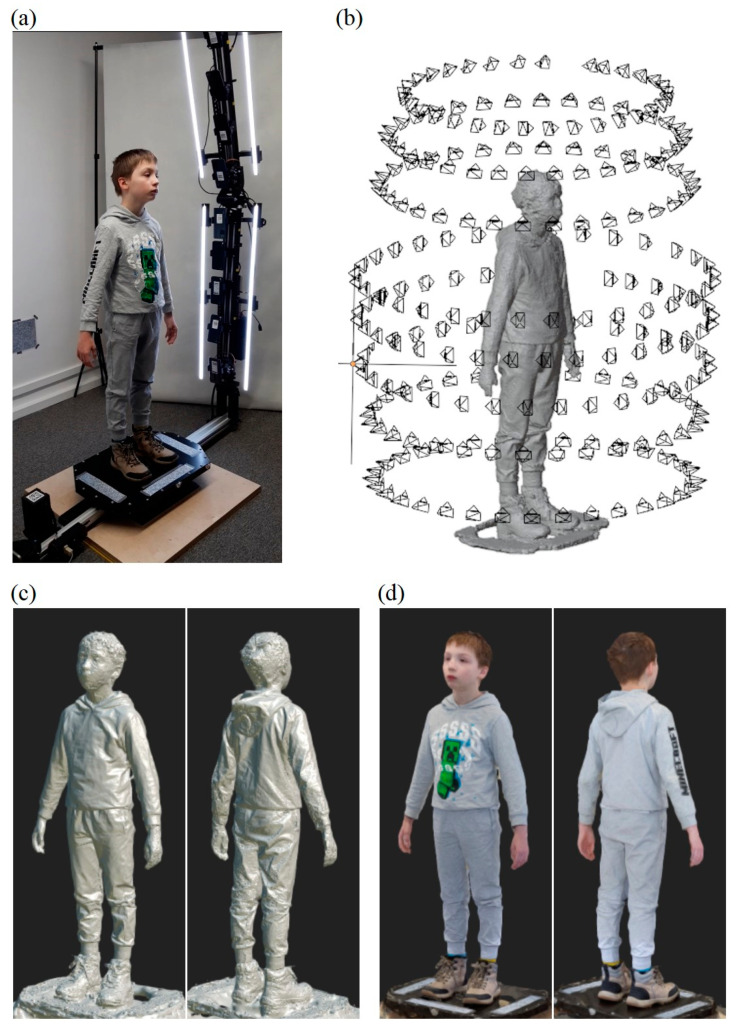
Scanning of a 1.4 m tall boy using HUBO scanner (**a**), determined camera poses (**b**), non-textured solid model (**c**), textured solid model (**d**).

**Table 1 sensors-23-05840-t001:** The configurations of analyzed variants of scanning and the results of quantitative research in terms of the RMSE quality rate for male and child dummies, respectively (*k*—number of cameras positions along the mast, *n*—number of cameras positions around the scanned object during one rotation of the mast, *p*—number of mast rotations for various options in terms of pitch angles of cameras, *c*—total number of camera poses).

Object	Male Dummy			Child Dummy		
Variant	b1m	d2m	e1m	b1c	d2c	e1c
*k* [-]	8	10	8	6	7	6
*n* [-]	32	32	32	32	32	32
*p* [-]	1	1	2	1	1	2
*c* [-]	256	320	512	192	224	384
RMSE [px]	0.796	0.867	0.944	0.860	0.847	0.980

## Data Availability

Not applicable.

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
