# Peer review of "Mechatronic Design and Experimental Research of an Automated Photogrammetry-Based Human Body Scanner"

_sensors, 2023, doi:10.3390/s23135840_

Round 1
Reviewer 2 Report
The paper presents a mechanical structure that should allow fastening one or more actual 3D scanning devices which are then expected to be turned around the subject in order reconstruct 3D human body. More specifically, instead of using the 3D scanner on the mechanical structure, the paper uses RGB camera from smartphones and subsequently process images acquired from them with ALICEVSION photogrammetry software. In that sense basically all the quantitative results are relevant only to the specific ALICEVSION software and settings used and can be hardly generalized to any another type of 3D scanner that would be put on the presented mechanical structure. More importantly human body scanning here lasts exceptionally long around 30 seconds, even when using 10 smartphones on it (!?). This seems to outweigh all the claimed advantages in the paper about used mechanical design. Since one can easily use a single smartphone, walk few times around the person of interest, and for even less than 30 seconds, and upload the images to ALICEVSION. In addition, there are human body scanners which scan from 3—4 views entire person in matter of seconds, if not even faster (e.g. https://www.indiamart.com/proddetail/body-scan-white-light-scanner-7422930230.html or https://www.youtube.com/watch?v=86hN0x9RycM ). In another words, scanning specifically humans can be done efficiently if some 3D scanner is put on 3-4 scanning poles around the person. No need for design and usage of the presented mechanical design. This led me to the conclusion that the presented solution makes little or no sense to be used, compared to the existing alternatives. Below are some specific comments related to the paper.
„at the same time allow faster and more accurate scanning than manual scanners.“ Manual scanners are not necessarily less accurate. Please elaborate this.
„it was assumed that the scanning time should be no longer than 3 minutes“ That is pretty long scanning time for a human to stand still.
Figure 1 is extremely unclear in terms of any technical details presented, about various parts. It seems like print/screen from some CAD software with extremely low image quality.
„Figure 1 15 – presence sensor“ What is that for?
Figure 1, is a sensor on the position 13 only or also on the position 12?
Section 2.6 “enabling the configuration of the working space of the device for the assumed dimensions of the object,” This is perhaps the most important, if not the only true advantage of this scanner since all other claimed contributions are present in many other commercial variants.
Section 3.1. Selection of parameters A-C are very much specific about what the actual scanner (e.g. cameras FOV, scanning accuracy, speed, sensitivity to object texture, surrounding light etc.) is used and makes no sense for a general case.
Provide some online video of the scanner in action, including the details when the scanner is put together and put apart for some transport, in order to better appreciate various details on construction design.
Section 3.2 This is qualitative analyses between ALICE vison SW (that is apparently used to process smartphones photos put on the mechanical arms of the presented structure) and Hexagon Stereo Scan (AICON) scanner. Therefore, this evaluation is quite meaningless if any other pairs of scanner have been used instead. Similarly Figure 9 showing theoretical number of cameras needed is absolutely irrelevant if for example structured light scanner would been used: in that case 1-2 position would normally suffice to capture entire person height. Moreover, if many other types of 3D scanners were used and placed on the lower and upper part of the presented mechanical construction (instead of just RGB cameras on two smartphones) some type of calibration between lower and upper scanner would be needed and/or 3D point cloud registration. Apparently, the presented mechanical design does not address this problem at all?
Section 4. using 10 cameras (smartphones) to capture a person of 1.82height is a huge overkill which largely overcomes all the claimed advantages of the presented solution (scalability, easy of usage, low cost, practical design etc.). I do not recall any use friendly human body scanner that would require capturing human with 10 different physical sensors, basically from one spatial pose (‘view’) only.
NA
Reviewer 3 Report
In general, the text is prepared on a current topic. However, the manuscript contains such long and unnecessary details. This makes it difficult to read and understand. It is not understood exactly how an original scientific contribution was made. The entire manuscript should definitely be shortened. It should be rewritten with shorter and clearer expressions. In addition to this, here are my recommendations:
1) The results of the study should be briefly mentioned in the abstract.
2)The introduction is too long. The problem definition can be made more concisely and clearly. In addition, a separate title should be created for Literature review.
3) It's quite long in Section 2. The manuscript may not give as detailed information as the HUBO scanning system manual. Please also introduce the system with its main lines.
4) The experimental part of the study should be developed. It is not fully understood on what basis the accuracy analysis is made. Can metrics other than RMSE be used for evaluation?
Moderate editing of English language
Round 2
Reviewer 2 Report
The paper is now ready to be published.
Reviewer 3 Report
I thank the authors for improving the manuscript in accordance with the recommendations. I suggest that the manuscript as such is acceptable.